

# Updates of C-LSAT 2.1 and the development of high-
# resolution LSAT and DTR datasets
Sihao Wei[1,2], Qingxiang Li[1,2,3], Qiya Xu[1,2], Zicheng Li[1,2], Hanyu Zhang[1,2], Jiaxue Lin[1,2]
*[1] School of Atmospheric Sciences, Sun Yat-sen University, and Key Laboratory of Tropical Atmosphere–Ocean*
*System, Ministry of Education, Zhuhai 519082, China*
*[2] Southern Laboratory of Ocean Science and Engineering (Guangdong Zhuhai), Zhuhai, China*
*[3] Research Center for Ecology and Environment of Central Asia, Chinese Academy of Sciences, Wulumuqi 830011,*
*China*
**Correspondence:** Qingxiang Li (liqingx5@mail.sysu.edu.cn)





**Abstract.** High-resolution climate datasets are of critical importance for the comprehension of spatial and temporal variations in climate and hydrology. However, their development is significantly influenced by the availability, density, and quality of observational data. Using the China global Land Surface Air Temperature 2.0 (C-LSAT 2.0) station data as a foundation, we collected and integrated nearly 3000 additional station observations and conducted the quality control and homogenization processing to complete the update of the C-LSAT 2.1 dataset. The coverage of Tavg, Tmax, and Tmin in the C-LSAT 2.1 dataset has been significantly enhanced, further enhancing the representativeness of global land diurnal temperature range (DTR) data with greater spatial heterogeneity. Compared to C-LSAT 2.0, C-LSAT 2.1 shows consistent overall trends, except for a slight increase in LSAT anomaly observed in the Southern Hemisphere after 2010. Furthermore, we employ a "Thin Plate Spline (climatology) + Adjust Inverse Distance Weighted (anomaly fields)" technical framework to develop a high-resolution (0.5° × 0.5°) LSAT (C-LSAT HRv1) and DTR (C-LDTR HRv1) dataset from January 1901 to December 2023. Except for some differences existing during the period of 1901–1950 due to the limited number of observational stations, the C-LSAT HRv1 and C-LDTR HRv1 datasets effectively capture the corresponding variation patterns at both global and regional scales for the other periods. The C-LSAT 2.1 dataset can be downloaded from https://doi.org/10.6084/m9.figshare.28255394.v1 (Wei et al., 2025a), while the C-LSAT HRv1 and C-LDTR HRv1 datasets are available at https://doi.org/10.6084/m9.figshare.28255505.v1 (Wei et al., 2025c) and https://doi.org/10.6084/m9.figshare.28255568.v1 (Wei et al., 2025b), respectively. These can also be accessed at http://www.gwpu.net (last accessed: December 2024).



## 1 Introduction

Global Surface Temperature (GST) is one of the most important elements in the Earth's climate system, it serves as a key indicator for monitoring and understanding climate change and directly reflects global warming (IPCC, 2007, 2013, 2021). Similarly, Land Surface Air Temperature (LSAT), which is closely related to GST, is also of critical importance. Since global industrialization, the rising emissions of greenhouse gases, such as carbon dioxide, have driven a rapid increase in LSAT, causing profound consequences on ecosystem stability, human health, and economic production (Jones et al., 2023; Loucks, 2021). The Intergovernmental Panel on Climate Change (IPCC) has systematically summarized and assessed climate change research through its assessment reports. These reports reveal the current state, future change, impacts, and adaptation measures of climate change, providing the scientific foundation for policy-making by governments worldwide. IPCC AR6 (2021) indicates that the global land temperature during 2011–2020 increased by 1.59 °C (1.34–1.83 °C) relative to pre-industrial levels.

The diurnal temperature range (DTR) indicates the difference between day and night temperatures, influenced by factors such as greenhouse gases, aerosols, and changes in land use (Kalnay and Cai, 2003; Stjern et al., 2020). DTR exhibits significant spatial heterogeneity and seasonal variations. In the latter half of the twentieth century, the increase in global land surface temperature at night was greater than during the day. This trend led to the narrowing of the global DTR (Zhong et al., 2023). Furthermore, the DTR change is strongly correlated with the probability of extreme high and low temperature events. According to IPCC AR6 (2021), global DTR has been decreasing since 1950, with the majority of the reduction occurring between 1960 and 1980.

Meteorological observation stations vary significantly in spatial distribution, particularly in high-altitude mountainous areas or regions with complex terrain. Additionally, disparities in temporal coverage and incomplete homogenization affect the accuracy of climate change analysis (Kumar et al., 2022; Sokol et al., 2021; Viviroli et al., 2011; Zhao et al., 2020). The major representative LSAT benchmark observational datasets worldwide used in IPCC AR6 include the CRUTEM (Osborn et al., 2021), GHCN (Menne et al., 2018), GISTEMP (Lenssen et al., 2024), Berkeley Earth (Rohde and Hausfather, 2020) and C-LSAT (Li et al., 2021; Sun et al., 2021), etc. Global land DTR datasets comprise CRU TS (Harris et al., 2020), GHCNDEX (Menne et al., 2018) and the recently released C-LDTR (Xu et al., 2025), etc. Some datasets





provide Tmax and Tmin, enabling the calculation of DTR, such as Berkeley Earth
(Rohde and Hausfather, 2020), HadEX3 (Dunn et al., 2024), and HadGHCND (Caesar
et al., 2006).

Improving spatial resolution is essential for investigating regional climate change,

especially in quantifying the effects of topography and supporting climate research at
medium and small scales, which can provide more accurate support for climate
prediction, regional model refinement, and climate risk evaluation (Beck et al., 2018;
Harris et al., 2014, 2020; Kotlarski et al., 2014; Sun et al., 2018). Global high-resolution
LSAT datasets have been continuously developed in recent years. However, they remain
constrained in capturing climate change in some regions (Karger et al., 2017; Li et al.,
2021; Wang et al., 2024; Li B et al., 2024). Therefore, it is essential to systematically
integrate supplementary observational networks to enhance the accuracy of datasets and
their capacity to capture climate change, especially at regional scales (Haylock et al.,
2008; Li et al., 2017, 2020; Menne et al., 2012; Wu and Gao, 2013; Xu et al., 2013).
Long-term series datasets are conventionally generated by separately interpolating the
climatology field and the anomaly field, and then combining them into a complete
dataset (Cheng et al., 2020; Harris et al., 2020; New et al., 1999, 2000; Schamm et al.,
2014). For climatology field interpolation, common methods include the Thin Plate
Spline (TPS) method (Wahba, 1990), Precipitation-elevation Regressions on
Independent Slopes Model (PRISM) method (Daly et al., 1994), and the Kriging
method (Cressie, 1990). When interpolating the anomaly field, the Inverse Distance
Weighted (IDW) method, Multiple Regression method, and Bilinear Interpolation
method are frequently employed. Among the above mentioned datasets, the Climatic
Research Unit (CRU) developed a 0.5° × 0.5° high-resolution global LSAT dataset by
interpolating the climatology field and anomaly field using the TPS method and
Angular Distance Weighting (ADW) method (New et al., 1999, 2000). The Berkeley
Earth team employed the Kriging method and IDW method to construct a high-
resolution global LSAT dataset with a 1° × 1° resolution (Rohde et al., 2013). Fick et
al. (2017) developed a global 1km LSAT dataset through application of the TPS method.

The C-LSAT dataset integrates observational datasets from over ten global,

regional, and national sources, continuously improving data completeness and accuracy
(Li, 2019; Li et al., 2021; Li Z 2023, 2024b; Sun et al., 2021, 2022; Sun and Li, 2021a,
b; Xu et al., 2018; Xu Q 2024, 2025; Yun et al., 2019). Currently, the C-LSAT group
only provides datasets at 5° × 5° resolution (C-LSAT 2.0, including Tavg, Tmax, and





Tmin) (http://www.gwpu.net) and recently released C-LDTR (Xu et al., 2025). This
study aims to utilize the recently updated C-LSAT 2.1 station data for updating the C-
LSAT 2.1 (5° × 5°) gridded data (Wei et al., 2025a), and to develop corresponding
global high-resolution LSAT (C-LSAT HR) and DTR (C-LDTR HR) datasets at a 0.5°
× 0.5° resolution (Wei et al., 2025b, c). Consequently, this study is organized into seven
main sections. Section 2 details the updates and pre-processing of the C-LSAT 2.1
station data. Section 3 introduces the C-LSAT 2.1 update (5° × 5°). The development
and validation of the C-LSAT HRv1 and C-LDTR HRv1 datasets are presented in Sect.
4. Section 5 analyzes the spatiotemporal patterns of global and regional LSAT and DTR
using high-resolution datasets (0.5° × 0.5°). Section 6 discusses the availability of these
datasets. The concluding section summarizes the key findings of the study.

## 112 2 Update and pre-processing of C-LSAT 2.1 station data

### 113 2.1 Data sources and update

### 114 2.1.1 Data integration

This study utilizes C-LSAT 2.0 station data (Xu et al., 2018; Yun et al., 2019), combined
with additional station data integrated from various countries, regions, and global
sources, covering the period from 2013 to 2023. Compared to the C-LSAT 2.0 station
data, the C-LSAT 2.1 station data significantly increased the number of observation
stations (Tavg increased from 15936 to 25085 stations, Tmax from 13648 to 25086
stations, and Tmin from 13629 to 25083 stations, as shown in Fig. 1 of Xu et al.(2025)).
Various data sources commonly assign different station IDs to the same station.
Therefore, how to match the data from various sources with the corresponding stations
in the C-LSAT station data is a problem that requires urgent resolution. Typically, most
stations have a core five-digit ID. For example, the core ID for the "JAN MAYEN"
station is 01001. In the GSOD, it appears as 01001099999, in the CLIMATE Report as
01001, and in the C-LSAT station data as 601001001000. However, some stations don't
follow this principle, so we employ the station name or identify nearby stations to locate
the corresponding stations and complete the update. Notably, when the sequence of a
station is derived from multiple data sources, there may be homogenization
discrepancies, which necessitate applying calibration procedures for the specific station.

### 131 2.1.2 Eliminating Duplicate Stations

When updating data from multiple sources, duplicate stations are inevitable. They





primarily originate from different station IDs in the data sources referring to the same
station, or emerge through new duplicates produced during iterative updates of the C-
LSAT station data. Duplicate stations can affect the interpolation of both the
climatology field and anomaly field, causing deviations in the interpolation results. To
address this issue, it is essential to eliminate duplicate stations. The process initiates
with filtering the C-LSAT 2.1 station data to identify any duplicate stations.
Subsequently, the corresponding update sources and time series from nearby stations
are plotted for comparison. A reference station is selected based on exhibiting a longer
or more reliable data continuity. The data from the duplicate stations are selectively
merged with the reference station or retained unmodified, ensuring the retention of a
single representative station for each group of duplicates (Rennie et al., 2014; Xu et al.,
2018).

**2.1.3 Update of Climatology**
The Tavg variable contains climatology (1961–1990) in the C-LSAT 2.1 station
data including 13756 stations. Among these 11907 stations calculate Tavg using the
average of Tmax and Tmin. The remaining 1849 stations, which lacking Tmax or Tmin
data, are primarily derived from datasets such as CRUTEM4, HISTALP, and SCAR.
Compared to other datasets, the C-LSAT 2.1 station data demonstrates substantial
improvements in station coverage in multiple regions, especially in East Asia. Figure 1
illustrates the C-LSAT 2.1 station data updates, compared to C-LSAT 2.0 station data,
the number of stations has significantly increased for Tmax, Tmin, and Tavg,
particularly after the 1970s. These additional stations substantially expand spatial
coverage, thereby enhancing the accuracy of data and reducing uncertainty after
gridding.

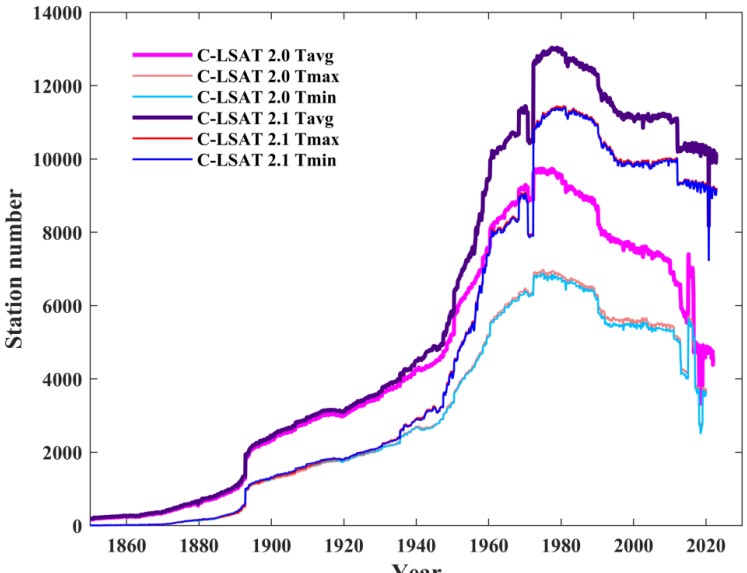

**Figure 1.** The update of C-LSAT 2.1 station data.

## 2.2 Data pre-processing

### 2.2.1 Quality control

Data quality control is a crucial step to ensure the accuracy and reliability of datasets. By identifying and eliminating outliers, invalid data, and measurement errors, this process reduces the influence of observational biases, ensuring the consistency and integrity of the data.

First, when updating station data, if a station has a data record exceeding 15 years, the newly updated data is subjected to this quality control process. Any anomaly—defined as the difference between the updated data and the previously averaged monthly data—that exceeds five times the standard deviation is classified as an outlier and will be treated as missing data.

Subsequently, when generating gridded data, we should do quality control on all station data. We follow the methods proposed by Lawrimore et al. (2011) and Menne et al. (2009) to implement the necessary quality control steps for the C-LSAT 2.1 station data. The results of the quality control process are shown in Table 1.

1. **Climatic outlier check:** Stations with monthly records exceeding 10 years were selected, with the period from 1961 to 1990 as the climatology. The



climatological mean value was subtracted from the selected stations to calculate anomalies for each station. The standard deviation (STD) for each month during the climatology period was subsequently calculated. Any data that exceeded five times the STD for the corresponding month was flagged as an outlier and excluded.

2. **Spatial consistency check:** Based on Equation (1), the anomaly data were evaluated by examining all stations. For each station i, all stations located within a 500 km radius were identified, up to a maximum of 20 neighboring stations (n≤20). The mean ($\bar{X}$) and standard deviation ($\sigma$) of the anomalies for these n+1 stations were calculated. If the absolute value of the difference between the value at station i and $\bar{X}$ exceeded three times the $\sigma$, this value was classified as an outlier and removed.

$$|X_i - \bar{X}| > 3\sigma \tag{1}$$

3. **Internal consistency check:** The Tmax, Tmin, and Tavg of station data were assessed. If Tavg was larger than Tmax or Tavg was smaller than Tmin, these values were identified as outliers and removed.

**Table 1.** Quality control results for C-LSAT 2.1 station data (unit: station month).

| Steps | Results of QC | |
|---|---|---|
| | Tavg | DTR |
| First step (check for outliers) | 19046(0.15%) | 19671(0.21%) |
| Second step (spatial consistency check) | 161753(1.31%) | 94022(0.99%) |
| Third step (internal consistency check) | 6469(0.05%) | 0(0%) |

**2.2.2 Homogenization**

Data homogenization is crucial for understanding climate change. Although its influence on a global or large scale may be limited, its impacts on local regions are often substantial (Peterson et al., 1998; Ribeiro et al., 2016). It removes data discontinuities caused by non-climatic factors such as station relocations, instrument changes, and environmental transformations (e.g., urbanization), ensuring that the data accurately reflects signals of climate change (Eccel et al., 2012; Jiao et al., 2023). Homogenized data enhances reliability and reduces the influence of errors.

The homogenization process of C-LSAT station data follows the work of Xu et al. (2025). Using the method proposed by Peterson and Easterling (1994), a reference series was constructed by selecting 3–5 neighboring stations with correlation



coefficients greater than 0.8 relative to the target station. Based on the spatial distances
of these stations, a reference LSAT series was generated through a weighted average of
first-order differences. Subsequently, the RHTest V4 software was used to detect and
correct discontinuities in the target series (Wang and Feng, 2010). The PMTred
algorithm (derived from the Penalized Maximal t-test, PMT) in RHTest V4 served as
the primary algorithm to detect discontinuities in the target station's monthly average
Tmax and Tmin series at a significance level of 5%. For any confirmed breakpoints, the
differences between the target series and the reference series were uniformly allocated
using the mean adjustment (Wang, 2008a, b). According to this procedure, we adjusted
726 breakpoints (in 420 stations) for the 25086 Tmax stations and 1276 breakpoints (in
754 stations) for the 25083 Tmin stations of the C-LSAT station data. The homogenized
Tmax and Tmin data were then combined into the LSAT and DTR datasets (Table 2).
**Table 2.** The number of breakpoints adjusted at each step of homogenization.

| Breaks | Tmax | Tmin |
|---|---|---|
| One | 244 | 440 |
| Two | 106 | 195 |
| Three | 48 | 67 |
| Four or more | 22 | 52 |
| Total breaks | 726 | 1276 |
| Total adjusted stations | 420 | 754 |
| Total stations | 25086 | 25083 |

## 217  3 Update of C-LSAT 2.1

Based on the C-LSAT 2.1 station data, we first applied the Climate Anomaly Method
(CAM) for gridding, and reconstructed the gridded data with high and low-frequency
component decomposition and empirical orthogonal telecorrelation (EOT)
reconstruction methods (Sun et al., 2021), which significantly enhancing the coverage
of early-period grid data. Subsequently, observational constraints were applied to
increase the reliability of the data, ultimately resulting in a high-coverage, high-
accuracy C-LSAT 2.1 dataset (5° × 5°).
Figure 2 shows a comparison of the LSAT anomaly time series among the updated
C-LSAT 2.1, C-LSAT 2.0, and other LSAT datasets, covering the global, Northern
Hemisphere, and Southern Hemisphere regions. C-LSAT 2.1 exhibits strong



consistency with other LSAT datasets in the long-term trend, with all showing a
significant warming trend, especially the accelerated warming since the 1970s. The
warming rates of C-LSAT 2.0 are 0.133±0.014, 0.145±0.016, and 0.098±0.011 °C
decade$^{-1}$ for the global, Northern Hemisphere, and Southern Hemisphere, respectively,
whereas C-LSAT 2.1 shows rates of 0.131±0.015, 0.141±0.017, and 0.101±0.011 °C
decade$^{-1}$. In C-LSAT 2.1, the warming rates in the global, Northern Hemisphere, and
Southern Hemisphere present slight changes. C-LSAT 2.1 has made optimization
adjustments over version 2.0. For the global, Northern Hemisphere, and Southern
Hemisphere, C-LSAT 2.1 is higher than C-LSAT 2.0 both before 1950 and after 2000
(particularly pronounced in the Southern Hemisphere). The increase before 1950 is
primarily driven by improved data coverage, while changes in other periods may stem
from our eliminating duplication process and updates to new data sources. These results
suggest that C-LSAT 2.1 more accurately reflects the trends in LSAT changes.

Open Access  Earth System  Discussions
Science
Data

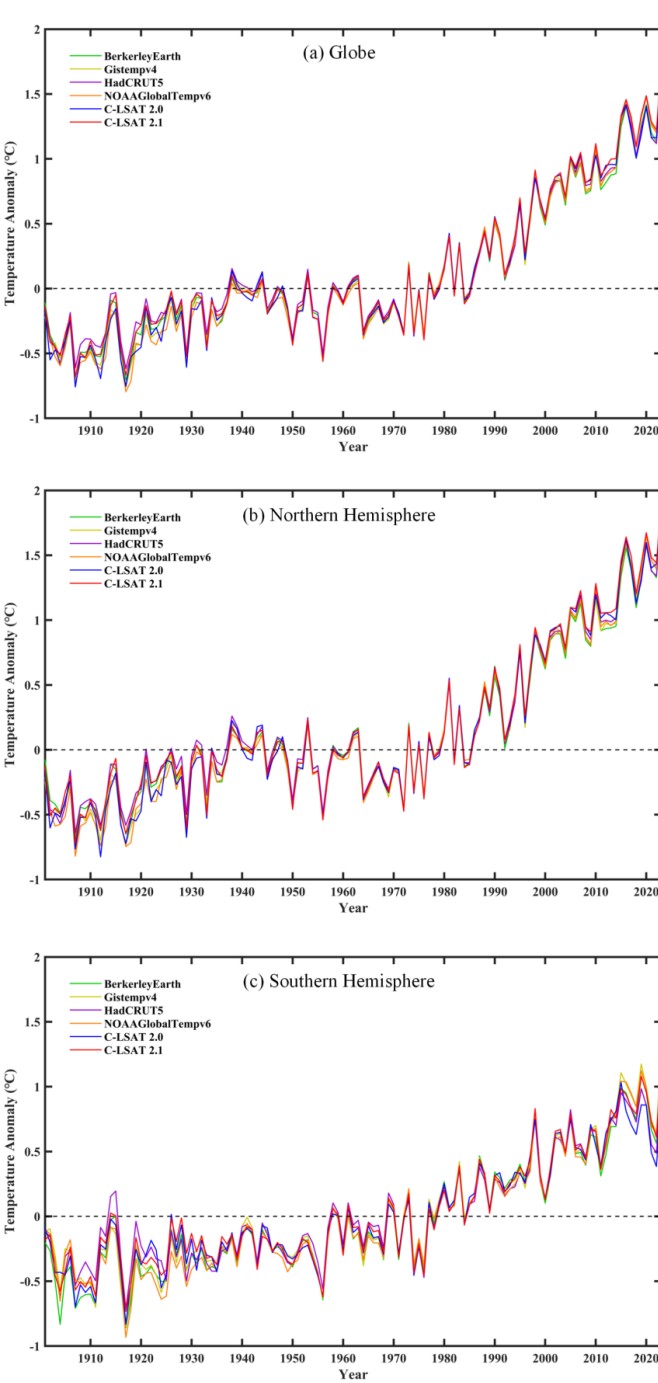


**Figure 2.** LSAT anomaly of C-LSAT 2.1 and other datasets from 1901 to 2023.



## 4 Development of C-LSAT HRv1 and C-LDTR HRv1

Building upon Cheng et al. (2020), this study also uses the TPS and Adjusted Inverse Distance Weighted (AIDW) methods to interpolate the climatology field and anomaly field of the C-LSAT 2.1 station data, ultimately generating the C-LSAT HRv1 and C-LDTR HRv1 datasets with a resolution of 0.5° × 0.5°.

### 4.1 Interpolation and validation of the climatology field

### 4.1.1 Interpolation and region division

This study employs the TPS method to interpolate the climatology field (1961–1990) of LSAT and DTR. The TPS method was initially proposed by Wahba (1990) and later optimized and improved by Hutchinson et al. (Hutchinson, 1998a, 1991, 1995, 1998b; Hutchinson and Gessler, 1994), evolving into the partial TPS method, which integrates covariate-dependent interpolation, extending the previous method that was limited to calculations based on independent variables. Based on the TPS method, Hutchinson et al. designed and developed the software ANUSPLIN, which enables multivariable data interpolation. This software has been widely adopted for meteorological data interpolation. The interpolation conducted in this study relies on it.

Due to the strong correlation between temperature and elevation, we selected longitude, latitude, and elevation as variables for interpolating LSAT and DTR. The elevation data used in this study was obtained from the ETOPO2022 published by NOAA (National Oceanic and Atmospheric Administration) (available at https://www.ncei.noaa.gov/products/etopo-global-relief-model). This dataset integrates topography, bathymetry, and coastline data from regional and global datasets, providing a comprehensive and high-resolution representation of the Earth's geophysical features.

Due to the Earth's spherical shape, the TPS method is unable to achieve a unified fit for the entire globe. Therefore, we must divide the globe into regions for separate interpolation. This study draws on the global partitioning scheme from the CRU (New et al., 1999) and WorldClim2 (Fick and Hijmans, 2017) datasets, dividing the globe into 20 regions for interpolation. The spatial distribution is shown in Fig. 3. In terms of station density, the highest density is observed around 40° N and 40° S, while the lowest density occurs at the poles and the equator. After interpolating the data for each region, the data from the 20 regions are merged into the global dataset. Nevertheless, one issue encountered is that when using ANUSPLIN to interpolate each region, the errors at the boundaries are typically larger. To address this, when interpolating the 20 regions, the

boundaries of each region are extended (by 5° latitudinally and 10° longitudinally).
After interpolation, the extended areas are clipped, and the data are then merged into
the global dataset. This approach effectively minimizes errors in the dataset.

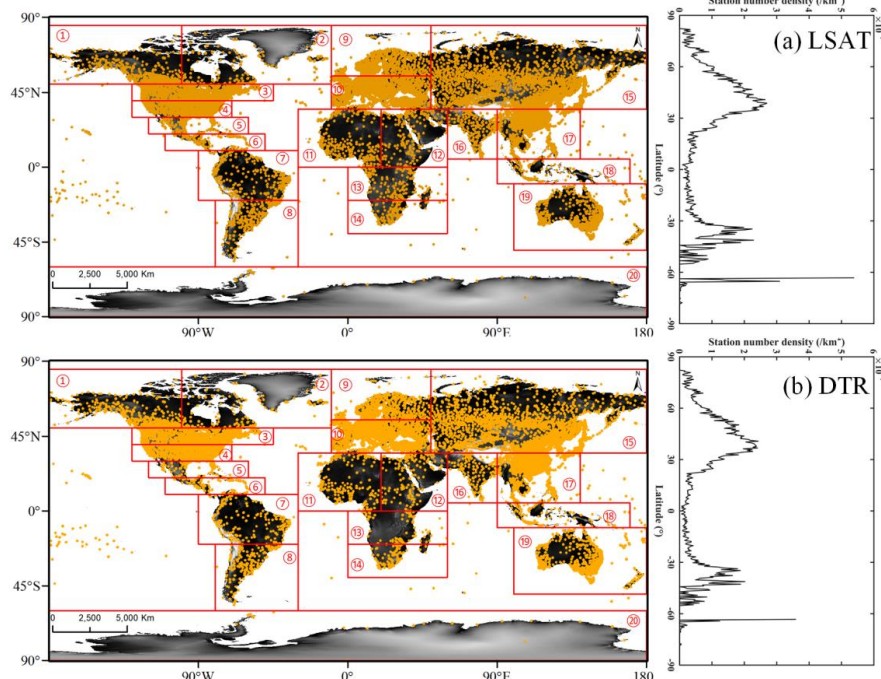


**Figure 3.** Spatial distribution of global LSAT (a), DTR (b) meteorological
observational stations and the division of 20 global regions.
**4.1.2 Validation of the climatology field**
When interpolating meteorological variables, we typically set longitude and
latitude as independent variables. However, whether elevation should be treated as an
independent variable or a covariate demands careful evaluation. There are three main
indicators for evaluating the interpolation accuracy of the climatology field: the square
root of generalized cross-validation (RTGCV), mean square residual (RTMSR), and the
data error variance estimate (RTVAR). RTGCV quantifies the overall error of data
fitting during the cross-validation process, measuring the model's generalization
capability. RTMSR reflects how well the model fits the input data, and RTVAR
evaluates the uncertainty in the data. Another indicator, Signal to Noise Ratio (SNR),
is typically used to indicate the complexity of the fitted surface. It represents the ratio



between the Signal and the Error value in the ANUSPLIN software output file. This
value generally needs to be less than 1 to indicate that the chosen interpolation scheme
is feasible.
The parameter schemes are detailed in Table 3, and the results are illustrated in Fig.
4–5. The overall error for DTR is higher than for LSAT. Experimental results revealed
that the interpolation error exhibited a marked increase when the spline order was set
to 4, compared with orders of 2 and 3. As a result, schemes A3 and B3 were excluded.
In the Antarctic (region 20), the 4 indicators of LSAT demonstrated substantial
increases, indicating that our data exhibit a large error in this area. Moreover, during
interpolation in the Antarctic, we found that the station density is notably low and
unevenly distributed. Considering the increased error mentioned before, both LSAT and
DTR for the Antarctic were excluded from this study. Future research will conduct a
more detailed and comprehensive investigation of the data in the Antarctic. Thus, the
subsequent contents of this study exclude the Antarctic (region 20). Following the
exclusion of the Antarctic, we compared the SNR, RTGCV, RTMSR, and RTVAR for
the remaining 19 regions. It was found that both LSAT and DTR attained the best results
with scheme B2 (Table 4). We adopted this scheme for interpolating the climatology
fields of LSAT and DTR.
**Table 3.** Climatology field interpolation schemes.

| Experiments | Independent spline variables | Covariates | Order of spline |
|---|---|---|---|
| A1 | Lat, Lon | Ele | 2 |
| A2 | Lat, Lon | Ele | 3 |
| A3 | Lat, Lon | Ele | 4 |
| B1 | Lat, Lon, Ele | / | 2 |
| B2 | Lat, Lon, Ele | / | 3 |
| B3 | Lat, Lon, Ele | / | 4 |






**Table 4.** Results of the climatology field interpolation schemes.

| Variables | Experiments | SNR | RTGCV | RTMSR | RTVAR |
|---|---|---|---|---|---|
| LSAT | A1 | 0.41 | 0.98 | 0.70 | 0.82 |
| | A2 | 0.28 | 1.00 | 0.79 | 0.89 |
| | A3 | 0.21 | 1.05 | 0.88 | 0.96 |
| | B1 | 0.27 | 0.98 | 0.77 | 0.87 |
| | B2 | 0.36 | 0.91 | 0.68 | 0.78 |
| | B3 | 0.35 | 0.91 | 0.68 | 0.78 |
| DTR | A1 | 0.37 | 1.65 | 1.23 | 1.42 |
| | A2 | 0.33 | 1.67 | 1.28 | 1.45 |
| | A3 | 0.23 | 1.72 | 1.43 | 1.57 |
| | B1 | 0.42 | 1.65 | 1.21 | 1.41 |
| | B2 | 0.36 | 1.62 | 1.22 | 1.40 |
| | B3 | 0.34 | 1.63 | 1.24 | 1.42 |

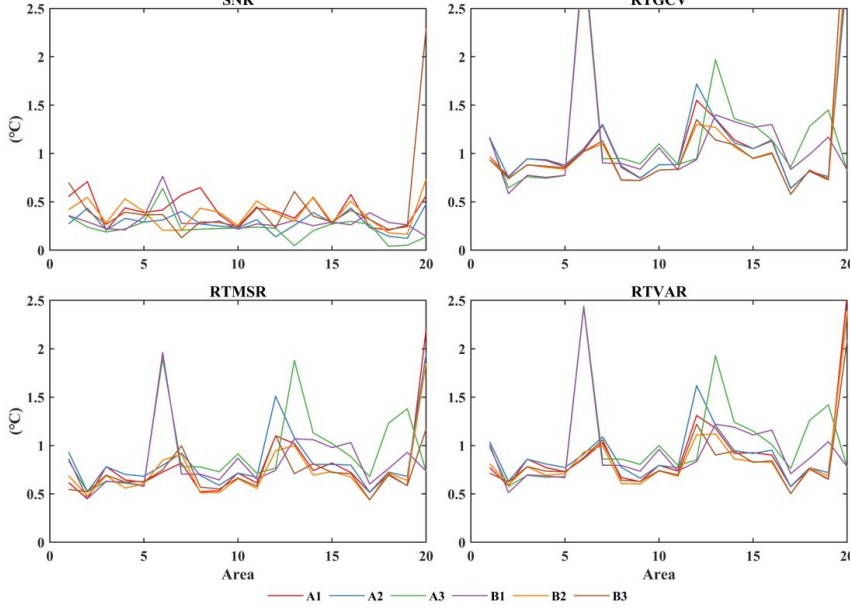


**Figure 4.** Cross-validation results of LSAT climatology field.



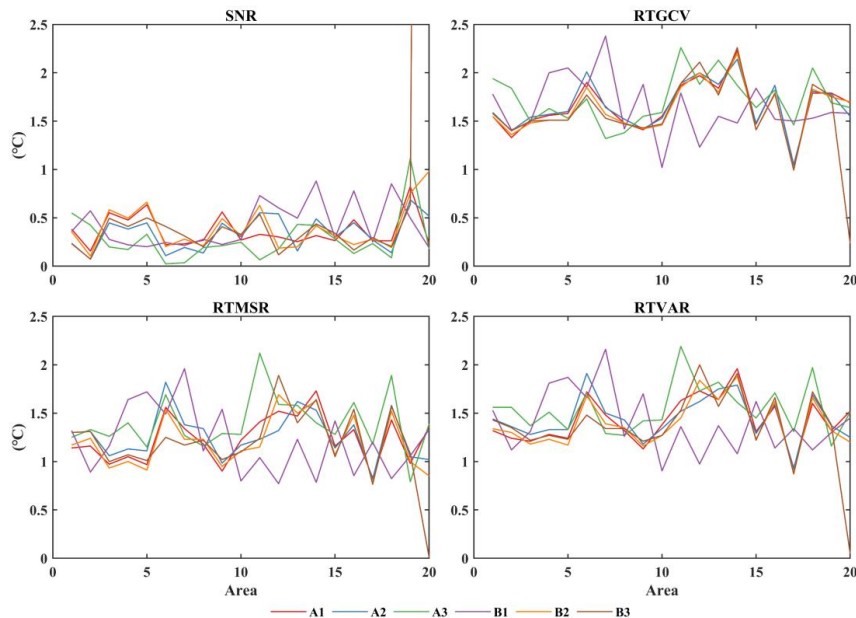


**Figure 5.** Cross-validation results of DTR climatology field.

Based on the cross-validation results, we evaluate the Mean Absolute Error (MAE)

and Root Mean Squared Error (RMSE) of the climatology fields for the C-LSAT HRv1
and C-LDTR HRv1 datasets (Fig. 6). For C-LSAT HRv1, the MAE and RMSE in the
Southern Hemisphere are smaller than the global average, whereas in the Northern
Hemisphere are greater than that in the global. In contrast to C-LSAT HRv1, the MAE
and RMSE of the C-LDTR HRv1 dataset show an opposite trend. The MAE and RMSE
reveal more significant asymmetries in both seasonal and regional performance, with
larger variability.

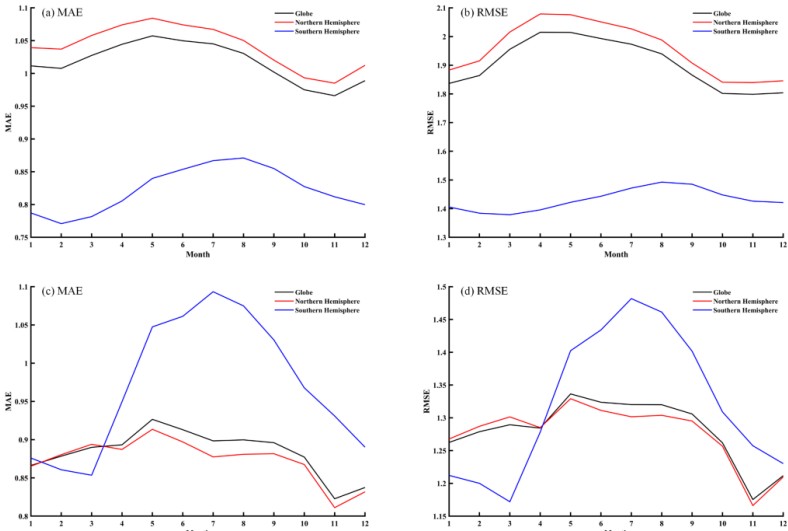

**Figure 6.** MAE and RMSE validation results of the climatology fields for C-LSAT
HRv1 (a–b) and C-LDTR HRv1 (c–d).

**4.2 Interpolation and validation of the anomaly field**

In this study, the Adjusted Inverse Distance Weighting (AIDW) method (Cheng et
al., 2020) was employed for spatial interpolation of the monthly anomalies from 1901
to 2023.

IDW assumes that spatially neighboring data points exhibit higher spatial
autocorrelation, and the closer a sample point is to the prediction point, the greater its
influence on the predicted value. The IDW method assigns weights to sample points
based on the inverse of the distance and then calculates the weighted average of the
values from each sample point. The equation is as follows:

$$T = \sum_{i=1}^{n} W_i T_i \tag{2}$$

$$W_i = \frac{d_i^{-\alpha}}{\sum_{i=1}^{n} d_i^{-\alpha}} \tag{3}$$

T represents the value at the prediction point, $T_i$ is the value at a given sample
point, $W_i$ is the weight of the sample point, n is the number of selected sample points,
$d_i$ is the distance from the sample point to the prediction point, and $\alpha$ is the parameter
that controls how the weight decays with distance. When using traditional IDW





interpolation, the weight exhibits rapid increase, even reaching infinity, as the distance
between two points approaches zero. This leads to the sample point having an
excessively high weight, which distorts the final estimated value. Building upon the
ADW method (New et al., 2000), this study modifies the weight calculation method of
the original IDW. The equation is as follows:
$$W_i = \frac{\left(e^{d_i/d_0}\right)^{-\alpha}}{\sum_{i=1}^{n}(e^{d_i/d_0})^{-\alpha}} \tag{4}$$

$d_0$ is the decay distance. Following the CRU05 (New et al., 2000), we adopted
values of 1200 km for LSAT interpolation and 750 km for DTR interpolation. Empirical
testing revealed that the optimal results were achieved when n = 6 and $\alpha$ = 4 (Cheng
et al., 2020). The AIDW method introduces an exponential decay relationship between
distance and weight, ensuring that the maximum weight does not exceed 1. The decay
curve is moderated, leading to a more reasonable distribution of weights.
After interpolating the anomaly fields of LSAT and DTR data, we analyze their
MAE and RMSE (Fig. 7). The results demonstrate that the trends of LSAT and DTR
exhibit strong coherence, both showing initial declines, reaching a minimum during the
1960–1990 period, and rebounding thereafter. This is strongly correlated with the
number of stations, and their trends are essentially opposite. The trend in the Northern
Hemisphere is largely consistent with the global trend. For LSAT, the Southern
Hemisphere is lower than the Northern Hemisphere and globe from 1901 to 1960, but
become slightly higher after 1960. Regarding DTR, the variability in MAE and RMSE
in the Southern Hemisphere are significantly higher than those in the Northern
Hemisphere and globe. During the 1901–1960 period, the three series are almost
identical, but after 1960, the MAE and RMSE in the Southern Hemisphere remain
consistently higher than those in the Northern Hemisphere and globe.

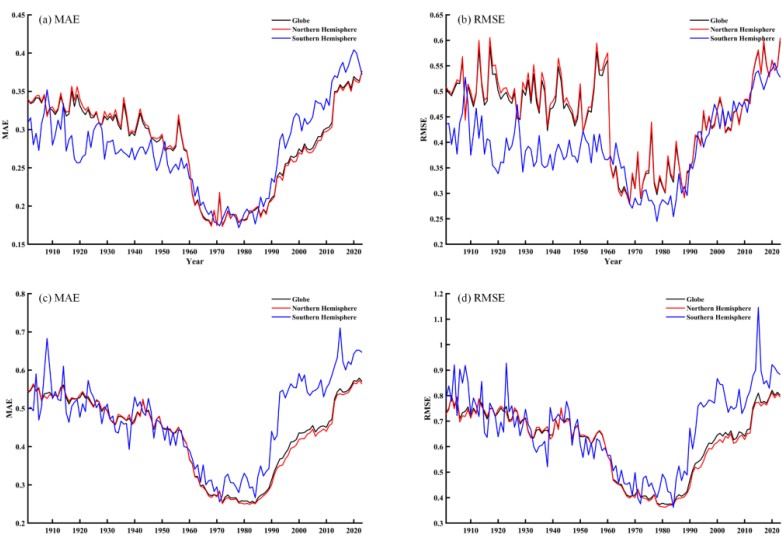


**Figure 7.** MAE and RMSE validation results of the anomaly fields for C-LSAT HRv1
(a–b) and C-LDTR HRv1 (c–d).

## 5 Spatiotemporal analysis of global LSAT and DTR

### 5.1 C-LSAT HRv1 climatology field

After interpolating the C-LSAT HRv1 climatology field, we assessed its performance
across the globe, Northern Hemisphere, and Southern Hemisphere. The highest LSAT
for the globe and Northern Hemisphere are observed in July, reaching 20.3 °C and
21.3 °C, respectively, while the lowest are recorded in January at 5.3 °C and -1.4 °C,
respectively. The Southern Hemisphere exhibits the opposite pattern, with the highest
and lowest LSAT observed in January (24.6 °C) and July (17.4 °C), respectively (Fig.
8). After excluding the Antarctic data, the Southern Hemisphere contains a smaller land
area, thus resulting in less influence on the global LSAT weight. As for the spatial
distribution, LSAT shows a dependency on both latitude and elevation, with
significantly lower in high-latitude regions (such as Northern North America and
Northern Asia) and high-elevation areas (such as the Tibetan Plateau and the Andes)
compared to other regions (Fig. 9).



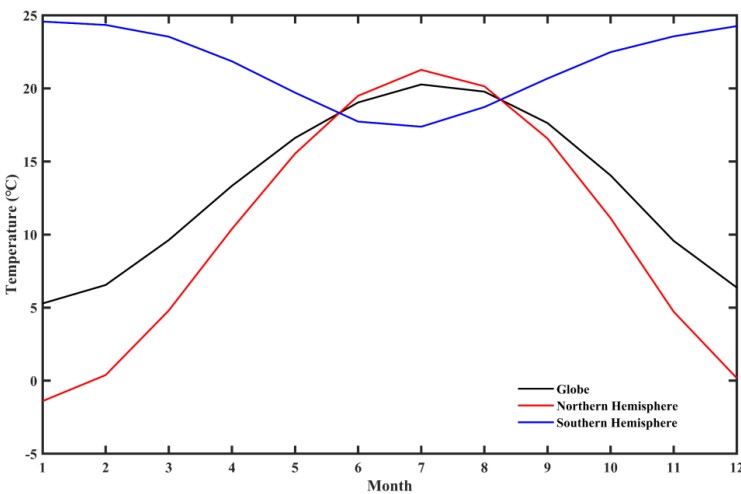

**Figure 8.** The LSAT for the C-LSAT HRv1 climatology field.

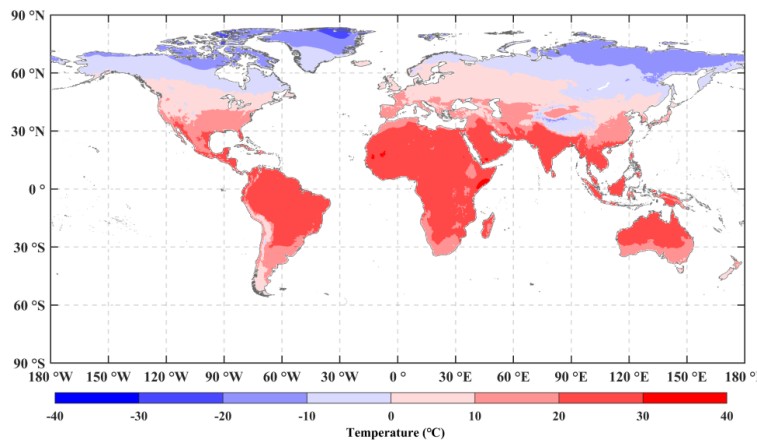

**Figure 9.** Spatial distribution of the LSAT for the C-LSAT HRv1 climatology field.

**5.2 C-LSAT HRv1 anomaly field**

**5.2.1 Global and hemispheric scales**

The LSAT anomaly variations of C-LSAT HRv1 and C-LSAT 2.1 from 1901 to 2023 for the globe, Northern Hemisphere, and Southern Hemisphere are presented in Fig. 10. The anomaly trends obtained in C-LSAT HRv1 are largely consistent with C-



LSAT 2.1, with warming rates of $0.131 \pm 0.015$, $0.140 \pm 0.017$, and $0.107 \pm 0.012$ °C
decade$^{-1}$ for the globe, Northern Hemisphere, and Southern Hemisphere, respectively.
The LSAT change trends for the globe and Northern Hemisphere demonstrate
comparable patterns, with warming predominantly concentrated in two periods: the
1900–1930s and the 1970–2020s, with accelerated warming in the latter period. A slight
cooling trend emerges in the middle period, from the 1940s to the 1960s. The warming
in the Southern Hemisphere is relatively slower and continues throughout the entire
1901–2023 period without experiencing the cooling trend observed in the global and
Northern Hemisphere during the 1940–1960s. Its warming rate also undergoes a
pronounced acceleration after the 1970s.
Table 5 presents the annual warming rates of the C-LSAT HRv1 for different
periods. The change is most gradual during 1901–1950, but after 1951, the warming
rate sharply increase, peaking in 1979, followed by a moderate decline in 1998. This
suggests that during the 1998–2014 hiatus, although no cooling is detected, the warming
rate is reduced.
Spatially, the LSAT across the globe, northern, and southern hemispheres show a
steady upward trend from 1901 to 2023, with recent years frequently establishing new
highest records for LSAT (with the Southern Hemisphere exhibiting a more gradual
increase). The LSAT change trend indicates continuous warming globally from 1901 to
2023, with the fastest warming occurring in regions such as Northern North America,
Eastern South America, Eastern Europe, and Eastern Asia (Fig. 11). Regarding different
periods, the fastest warming was observed between 1998–2023 (particularly in areas
north of 60° N), while the slowest warming occurred during 1901–1950 (Fig. 12).

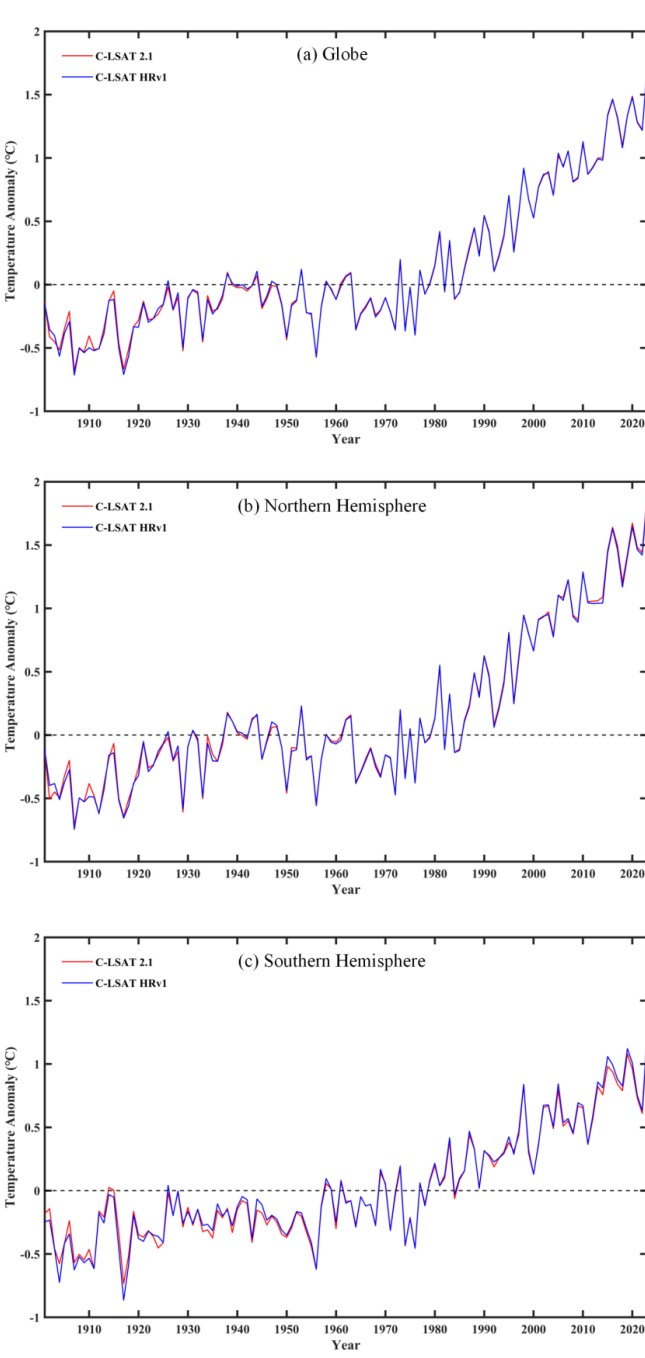


**Figure 10.** The LSAT anomaly in the globe (a), Northern Hemisphere (b), and
Southern Hemisphere (c) from 1901 to 2023 for both C-LSAT HRv1 and C-LSAT 2.1.



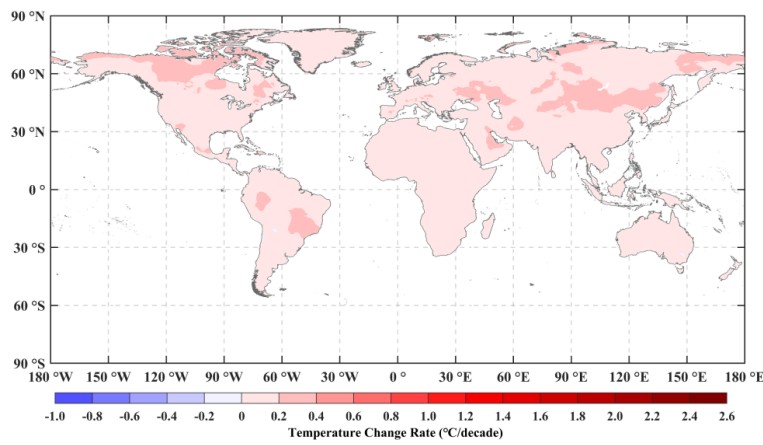


**Figure 11.** Spatial distribution of the LSAT change rate for the C-LSAT HRv1
anomaly field from 1901 to 2023.

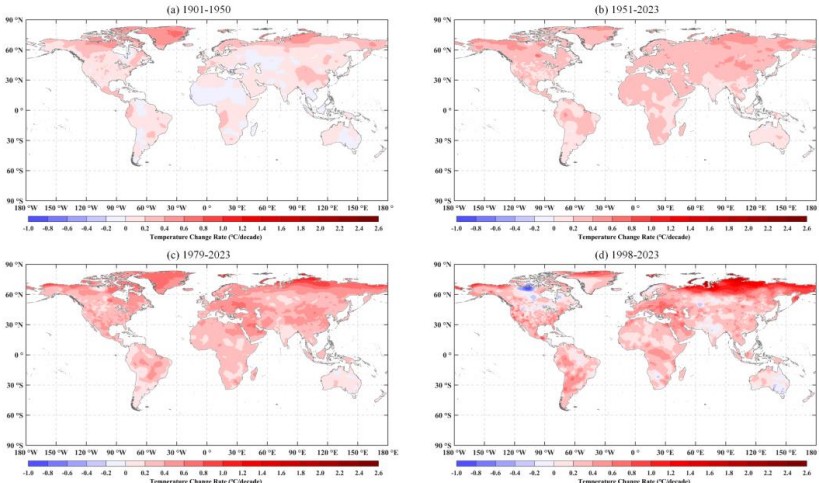


**Figure 12.** Spatial distribution of the LSAT change rates for the C-LSAT HRv1
anomaly field during 1901–1950 (a), 1951–2023 (b), 1979–2023 (c), and 1998–2023
(d).





**Table 5.** The LSAT change rates and their 95% confidence intervals for C-LSAT
HRv1 in the globe, Northern Hemisphere, and Southern Hemisphere over five
different periods (°C decade⁻¹).

|  | 1901–1950 | 1901–2023 | 1951–2023 | 1979–2023 | 1998–2023 |
|---|---|---|---|---|---|
| Globe | 0.096 ± 0.033* | 0.131 ± 0.015* | 0.243 ± 0.026* | 0.329 ± 0.041* | 0.303 ± 0.086* |
| Northern Hemisphere | 0.108 ± 0.037* | 0.140 ± 0.017* | 0.265 ± 0.030* | 0.371 ± 0.047* | 0.330 ± 0.091* |
| Southern Hemisphere | 0.063 ± 0.034* | 0.107 ± 0.012* | 0.179 ± 0.022* | 0.208 ± 0.041* | 0.228 ± 0.110* |

**5.2.2 Continental scale**
At the continental scale, both C-LSAT HRv1 and C-LSAT 2.1 show a warming
trend across all six continental domains since the early 20th century, with this trend
intensified after the 1970s and manifesting regional differences (Fig. 13). The warming
is pronounced in Asia, Europe, and North America, whereas it remains comparatively
moderated in South America, Africa, and Oceania, reflecting the different responses of
the climate system to global warming. Both datasets are consistent in their long-term
trends; however, differences in short-term fluctuations may stem from variations in
spatial resolution and processing methods.

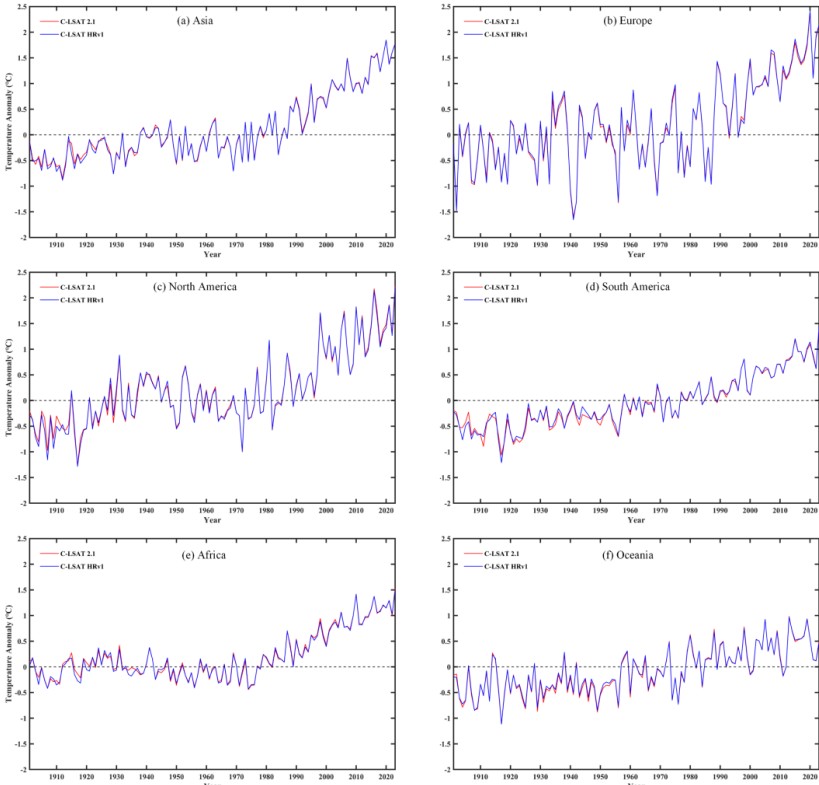


**Figure 13.** The LSAT anomaly for C-LSAT HRv1 and C-LSAT 2.1 in different
continents from 1901 to 2023.

**5.3 C-LDTR HRv1 climatology field**

Figure 14 shows that the monthly average DTR of the C-LDTR HRv1 climatology
field reverse in May for the globe, Northern Hemisphere, and Southern Hemisphere.
The global DTR reaches its maximum in April (11.8 °C) and attains its minimum in
December (10.8 °C). In the Northern Hemisphere, the DTR peaks in April (12.0 °C)
and reaches its minimum in November (10.6 °C), while in the Southern Hemisphere,
the peak occurs in August (13.2 °C) and the minimum in February (11.0 °C). The
Southern Hemisphere shows the largest DTR variation, significantly larger than that of
the global and Northern Hemispheres, primarily attributed to the smaller land area in
the Southern Hemisphere, resulting in higher sensitivity. This difference reflects the
combined impact of solar radiation, surface characteristics, and seasonal changes on the
climate system. Spatially, DTR depends not only on elevation but also is influenced by



land use and land cover in the region. DTR is higher in mountainous, plateau areas, and
deserts (Fig. 15).

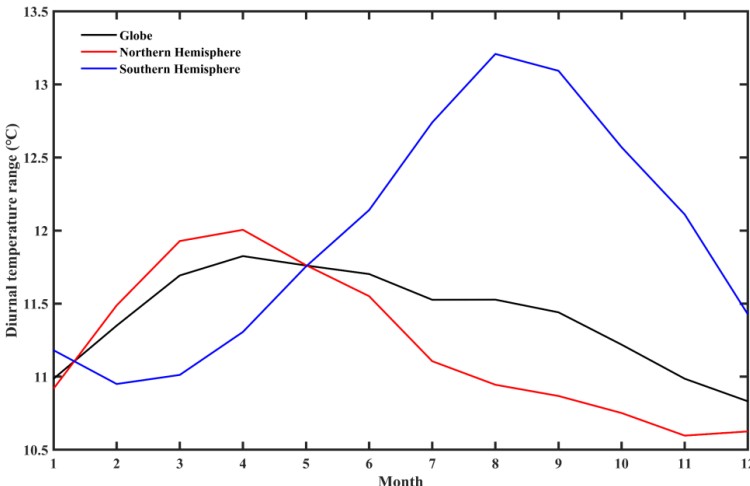


**Figure 14.** The DTR for the C-LDTR HRv1 climatology field.

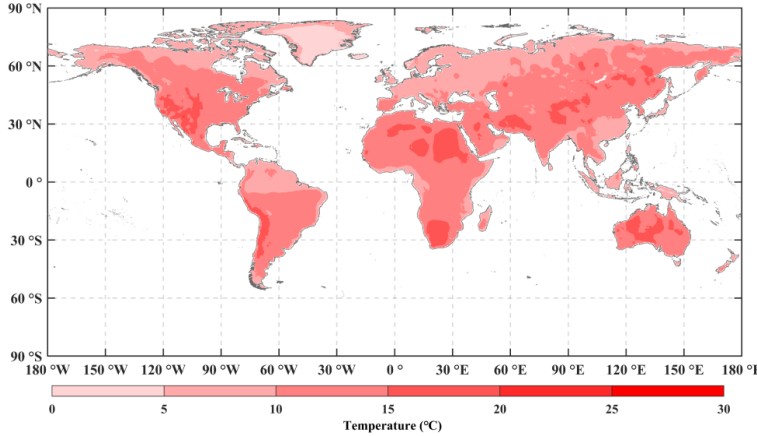


**Figure 15.** Spatial distribution of the DTR for the C-LDTR HRv1 climatology field.
**5.4 C-LDTR HRv1 anomaly field**
**5.4.1 Global and hemispheric scales**

The DTR anomaly changes of C-LDTR HRv1 for the globe, Northern Hemisphere,





and Southern Hemisphere from 1901 to 2023 are presented in Fig. 16. During 1950–
2010, C-LDTR HRv1 remains highly consistent with the C-LDTR, with change rates
of -0.031 ± 0.006, -0.038 ± 0.006, and -0.011 ± 0.011 °C decade$^{-1}$ for the globe,
Northern Hemisphere, and Southern Hemisphere, respectively. However, there are
notable discrepancies before 1950 and after 2010. From 1901 to 1950, the station
number is limited, leading to greater uncertainty, which is why the differences between
the two datasets are more pronounced. This is particularly apparent in the Southern
Hemisphere, where the DTR fluctuations and the differences between the two datasets
are significantly larger than those in the globe and Northern Hemisphere. After 2010,
the reduction in DTR (or Tmax and Tmin) station data lead to the differences between
C-LDTR HRv1 and C-LDTR, which is further reflected in other DTR datasets (Xu et
al., 2025). The DTR is stable during the 1900–1940s and 1980–1990s, declines during
the 1950–1970s, and shows a slight increase after the 2000s.
Table 6 shows the DTR change rates of C-LDTR HRv1 for different periods. The
change rate is stable from 1901 to 1950, then initiates a decline in 1951, stabilizes again
in 1979, and peaks at 1998. The DTR change rate in the Southern Hemisphere is more
pronounced than that in the globe and Northern Hemisphere.
It is noteworthy that there is no obvious spatial pattern in the changes in the DTR.
During the period of most significant change: 1998–2023, the regions with the most
rapid DTR increases are North America, Europe, and Oceania, whereas other regions,
including Africa, East Asia, South Asia, and the Middle East, demonstrate a pronounced
downward trend (Fig. 17–18).

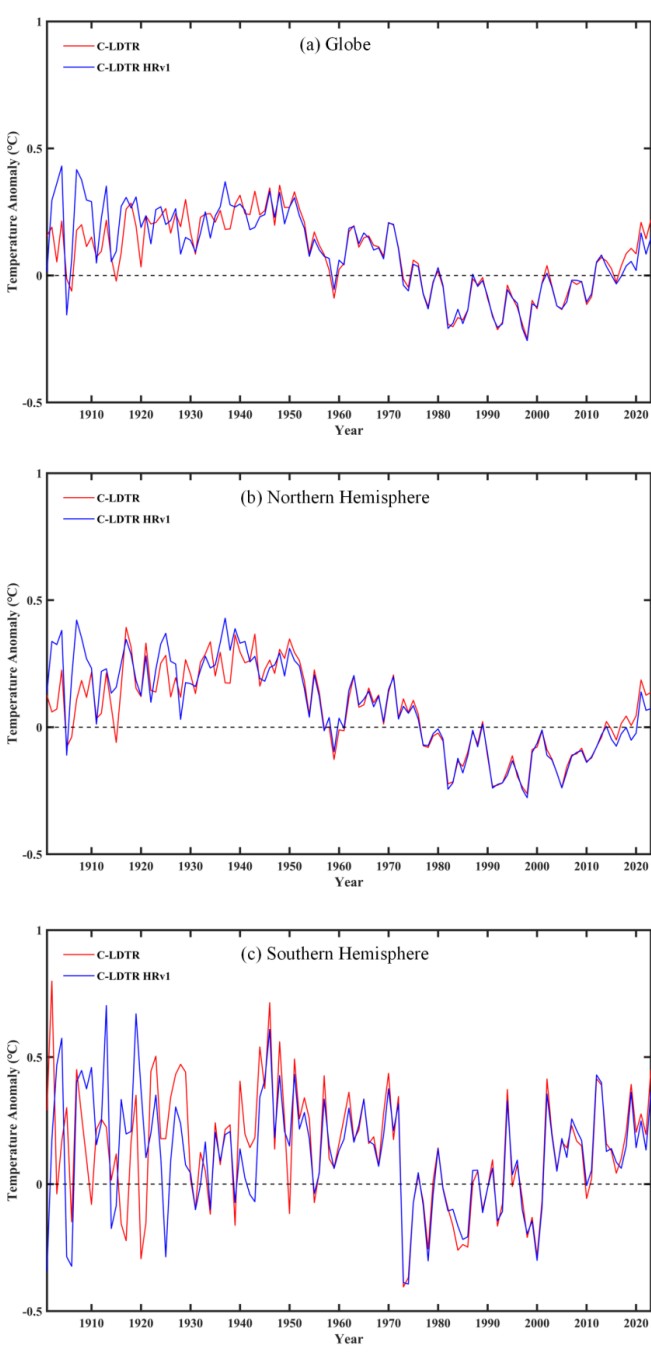


**Figure 16.** The DTR anomaly in the globe (a), Northern Hemisphere (b), and
Southern Hemisphere (c) from 1901 to 2023 for both C-LDTR HRv1 and C-LDTR.

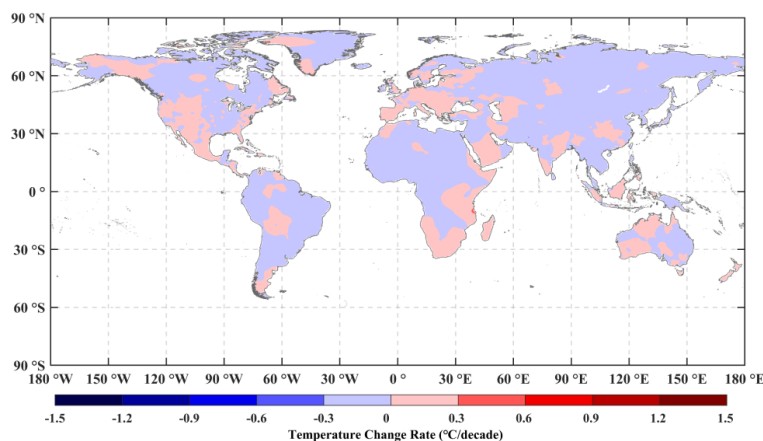


**Figure 17.** Spatial distribution of the DTR change rate for the C-LDTR HRv1
anomaly field from 1901 to 2023.

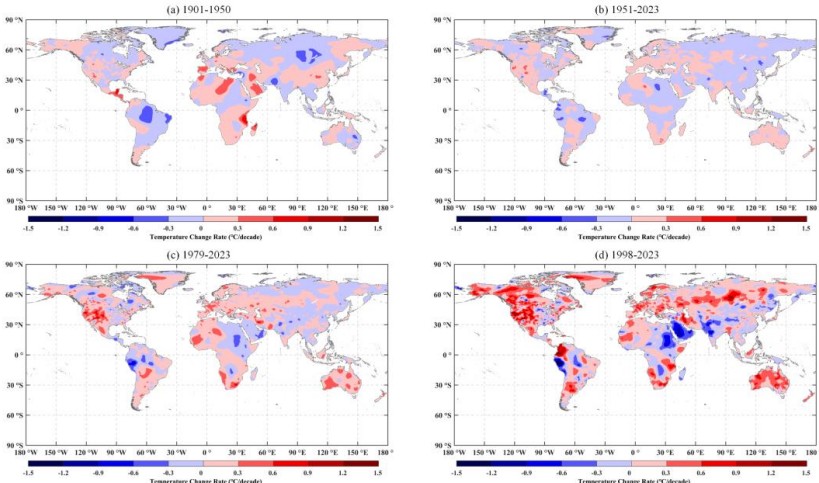


**Figure 18.** Spatial distribution of the DTR change rates for the C-LDTR HRv1
anomaly field during 1901–1950 (a), 1951–2023 (b), 1979–2023 (c), and 1998–2023
(d).



**Table 6.** The DTR change rates and their 95% confidence intervals for C-LDTR
HRv1 in the globe, Northern Hemisphere, and Southern Hemisphere over five
different periods (°C decade$^{-1}$).

|  | 1901–1950 | 1901–2023 | 1951–2023 | 1979–2023 | 1998–2023 |
|---|---|---|---|---|---|
| Globe | 0.007 ± 0.022 | -0.031 ± 0.006* | -0.023 ± 0.013* | 0.044 ± 0.018* | 0.097 ± 0.032* |
| Northern Hemisphere | 0.011 ± 0.020 | -0.038 ± 0.006* | -0.031 ± 0.013* | 0.032 ± 0.020* | 0.088 ± 0.035* |
| Southern Hemisphere | -0.004 ± 0.050 | -0.011 ± 0.011 | 0.001 ± 0.022 | 0.081 ± 0.034* | 0.124 ± 0.085* |

**5.4.2 Continental scale**
Based on the C-LDTR HRv1 and C-LDTR datasets, Fig. 19 illustrates the complex
variation characteristics and significant regional differences of DTR across six
continents between 1901 and 2023. DTR in Asia, Africa, and South America shows a
downward trend, whereas the changes in Europe, North America, and Oceania remain
relatively stable. Europe demonstrates a general upward trend throughout the entire
1901–2023 period, while DTR in the remaining five continents declines before the
1970s but rebounds after 2010.



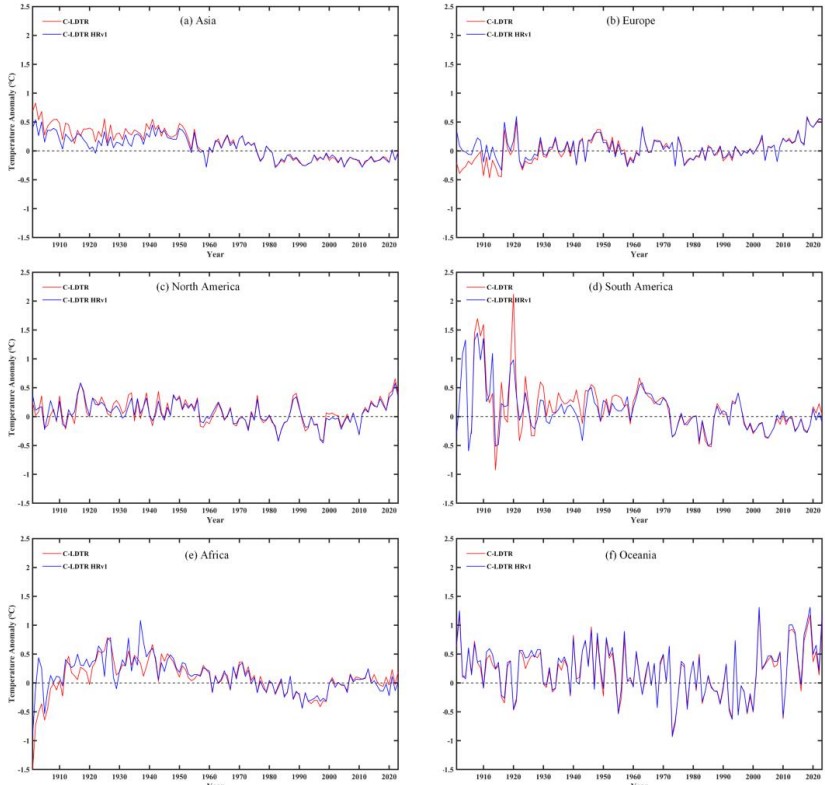

**Figure 19.** The DTR anomaly for C-LDTR HRv1 and C-LDTR in different continents from 1901 to 2023.

## 6 Data availability

The C-LSAT 2.1 dataset is publicly available on the website at https://doi.org/10.6084/m9.figshare.28255394.v1 (Wei et al., 2025a). The C-LSAT HRv1 can be downloaded at https://doi.org/10.6084/m9.figshare.28255505.v1 (Wei et al., 2025c). The C-LDTR HRv1 can be downloaded at https://doi.org/10.6084/m9.figshare.28255568.v1 (Wei et al., 2025b). They can also be accessed at http://www.gwpu.net (last accessed: December 2024) for free.

## 7 Conclusions

This study provides a comprehensive overview of the updates made to the C-LSAT 2.1 station data and grid data (5° × 5°). On this basis, the high-resolution (0.5° × 0.5°) LSAT



(C-LSAT HRv1) and DTR (C-LDTR HRv1) datasets are developed. The key
characteristics of the C-LSAT 2.1 station data, C-LSAT 2.1, C-LSAT HRv1, and C-
LDTR HRv1 datasets are summarized below:
1. C-LSAT 2.1 station data supplemented and integrated meteorological
observational data from various sources, resulting in a substantial enhancement in
global station coverage. After filtering based on the reference period (1961–1990), the
number of stations for LSAT and DTR is 13756 and 11907, respectively. The number
of stations peaks in the 1970–1980s, followed by a slight decline.
2. The updated station data was utilized for gridded interpolation and EOT
reconstruction (C-LSAT 2.1). Compared to C-LSAT 2.0, the LSAT change trends at the
global and hemispheric scales exhibit no significant change in C-LSAT 2.1.
3. Comparative analysis of C-LSAT HRv1 with other LSAT datasets. The results
show minor discrepancies in the period from 1901 to 1950, but the trends thereafter
demonstrate strong coherence. During the climatology period (1961–1990), the highest
LSAT in C-LSAT HRv1 are 20.3 °C (July) for the globe, 21.3 °C (July) for the Northern
Hemisphere, and 24.6 °C (January) for the Southern Hemisphere. The lowest LSAT are
5.3 °C (January) globally, -1.4 °C (January) in the Northern Hemisphere, and 17.4 °C
(July) in the Southern Hemisphere. The 1901–2023 warming rates for C-LSAT HRv1
are $0.131 \pm 0.015$ °C decade$^{-1}$ globally, $0.140 \pm 0.017$ °C decade$^{-1}$ for the Northern
Hemisphere, and $0.107 \pm 0.012$ °C decade$^{-1}$ for the Southern Hemisphere.
4. By comparing C-LDTR HRv1 with other DTR datasets, we find differences
between the datasets before 1950 and after 2010, with the former showing pronounced
discrepancies, especially in the Southern Hemisphere. Notably, strong consistency is
observed in other periods. The monthly variation of the DTR during the climatology
period differs significantly from LSAT, with the highest DTR reaching 11.8 °C (April)
globally, 12.0 °C (April) in the Northern Hemisphere, and 13.2 °C (August) in the
Southern Hemisphere. Whereas the lowest values are 10.8 °C (December) globally,
10.6 °C (November) in the Northern Hemisphere, and 11.0 °C (February) in the
Southern Hemisphere. Over the 1901–2023 period, the C-LDTR HRv1 shows the
change rates of $-0.031 \pm 0.006$ °C decade$^{-1}$ globally, $-0.038 \pm 0.006$ °C decade$^{-1}$ for the
Northern Hemisphere, and $-0.011 \pm 0.011$ °C decade$^{-1}$ for the Southern Hemisphere.
In summary, C-LSAT HRv1 maintains high consistency with other LSAT datasets.
In contrast, there are some differences between C-LDTR HRv1 and various DTR
datasets. Early-period discrepancies are primarily attributable to the limited number of



stations. The reduction in DTR (or Tmax and Tmin) station data lead to differences
between C-LDTR HRv1 and other DTR datasets in later periods.
**Author contributions.** SW: conceptualization, data curation, formal analysis,
investigation, methodology, resources, software, validation, visualization, Writing –
original draft preparation, writing - review & editing. QL: conceptualization, funding
acquisition, investigation, methodology, project administration, resources, software,
supervision, writing - review & editing. QX: data curation, formal analysis, resources,
visualization. ZL: data curation, formal analysis, resources. HZ: resources, validation.
JL: resources, validation.
**Competing interests.** At least one of the (co-)authors is a member of the editorial board
of Earth System Science Data. The authors have no other competing interests to declare.
**Acknowledgments.** This research was jointly supported by the Natural Science
Foundation of China (grant no. 42375022) and the National Key R&D Program of
China (grant no. 2023YFC3008002).
**Financial support.** This research was jointly supported by the Natural Science
Foundation of China (grant no. 42375022) and the National Key R&D Program of
China (grant no. 2023YFC3008002).

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
