# Peer review of "Updates of C-LSAT 2.1 and the development of high- # resolution LSAT and DTR datasets"

_Earth System Science Data, 2025_

## Author Comment (AC1)

This manuscript presents an important update and improvement based on a widely recognized dataset. I commend the authors for their excellent work. Building on the C-LSAT 2.0 dataset, the authors have integrated nearly 3,000 additional global observational stations to develop the C-LSAT 2.1, along with high-resolution (0.5°×0.5°) LSAT and DTR datasets. The new version significantly enhances data coverage, refines processing techniques, and provides valuable high-resolution products. The manuscript is comprehensive, logically structured, and the conclusions are sound. It fits well within the scope of Earth System Science Data (ESSD). However, the following minor revisions are recommended prior to publication:

**Response:** Thank you very much for your positive and constructive comments. We have tried our best to substantially modify the paper according to your review.

1. Please supplement the temporal coverage of observation stations at both annual and monthly scales.

**Response:** Thank you for your suggestion. We have now added a detailed summary of station temporal coverage at both annual and monthly scales. (Lines 150–153 and Table S1)

2. In section 4.1.2, the authors used the MAE, RMSE to evaluate the different datasets. However, sometimes, MAE and RMSE only provide one aspect performances of different datasets. DISO as a comprehensive performance evaluating index can illustrate the overall performance for all the datasets. Therefore, the DISO should be added in this study.

**Response:** We have now computed the DISO index for all datasets and included these results alongside MAE and RMSE in Section 4.1.2 and Section 4.1.3. (Lines 319–336, Line 376–379 and Figs. S1–S2)

3. If feasible, consider adding additional validation in representative regions, particularly in high-altitude or complex terrain areas.

**Response:** We have added validation for the Tibetan Plateau. (Lines 330–336, Line 378–379 and Figs. S1–S2)

4. All variables in equations should be consistently italicized.

**Response:** We have revised the equation descriptions to ensure that every variable is consistently set in italics.

5. Insert appropriate spaces between numbers, symbols, and units (e.g., in line 230).

**Response:** Thank you for highlighting this formatting issue. We have inserted appropriate spaces between numbers, symbols, and units throughout. (Lines 227–230)

6. Add a space before parentheses in Table 1 for clarity.

**Response:** We have modified it. (Table 1)

7. Please explain the observed increases in MAE and RMSE since 1990 as shown in Figure 7.

**Response:** This is due to the reduction in the number of stations after 1990. (Lines 366–369)

8. The abnormal warming pattern in northern North America depicted in Figure 12(d) requires

verification.

**Response:** We have examined the data in this region and corrected or removed some problematic records.

9. The colorbars in Figures 11 and 17 hinder regional comparisons; consider revising them for clarity.

**Response:** We have adjusted the colorbar of these figures.

10. There are citation errors that need correction (e.g., in line 776).

**Response:** We have carefully revised the references.

---

## Author Comment (AC2)

General comments

The manuscript describes the update of a valuable dataset of Land Surface Temperature through the inclusion of a significant number of additional stations, and the creation using the TSP method of high-resolution datasets of land surface temperature and diurnal temperature range. The manuscript is clear and scientifically sound. I have two main concerns that should be addressed before publication:

**Response:** Thanks for your positive evaluation and suggestion. We will do our best to address all suggestions and improve the manuscript.

1. It's not clear whether the trend uncertainties (e.g Line 230) have been computed taking the serial correlation of the time series into account, as the time series (e.g. Figure 2) seem to display a significant autocorrelation/serial dependence, which biases downwards the uncertainties if not accounted for. This is something that should be clarified.

**Response:** The serial correlation of the time series has been taken into account in the calculation of trend uncertainties, as referenced in Li et al., 2021. (Lines 230–231)

Li, Q., Sun, W., Yun, X., Huang, B., Dong, W., Wang, X. L., Zhai, P., and Jones, P.: An updated evaluation of the global mean land surface air temperature and surface temperature trends based on CLSAT and CMST, Clim. Dyn., 56, 635–650, https://doi.org/10.1007/s00382-020-05502-0, 2021.

2. The quality of the figures should be improved, to make them easier to understand, particularly in case of colour vision deficiencies.

**Response:** Thank you for your constructive comment. We have revised the figures in the manuscript.

Specific comments

1. Line 138: maybe give some more details on the filtering procedure

**Response:** The filtering procedure is mainly conducted based on the same core IDs and similar station names. We have supplemented the corresponding content accordingly. (Lines 134–135)

Figure 1: The colours in Figure 1 are very similar for some of the datasets and very difficult to distinguish – particularly for I suggest using more contrasting colours and different types of line (e.g. dashed, dotted) to make the figure clearer.
Response: We have modified it.

Line 166: maybe use : before "Any anomaly..." to make more obvious that the sentence is referring to the quality control process.
Response: Done.

4. The same notation could be used to denote standard deviation, STD is used in line 177, and sigma in line 184.

Response: We have modified it.

5. Table 1: the caption and text should be improved, to make clear what is exactly shown in the table – maybe instead of results of QC something like number of data values excluded during

the QC procedure? The "unit:station month" could also be made clearer. **Response:** Accepted and revised. (Lines 169–170)

6. Line 221: enhance (instead of " which significantly enhancing") **Response:** Accepted and revised.

7. Figure 2: same as for Figure 1, it's hard to distinguish the different colours / lines. **Response:** We have adjusted Figure 2.

8. Line 230: please indicate if autocorrelation was taken into account in the estimation of trend uncertainties.

Response: Done. (Lines 230–231)

9. Table: Ele was not defined for elevation in the text.Response: We have defined "Lat" "Lon" and "Ele" in the Table 1 caption for clarity. (Lines 311–312)

10. Figures 6, 7: the figure would be clearer with a slightly larger size of the text in the axis and legend; using different line styles (dashed, etc...) would make the figure easier to perceive in case of colour vision deficiencies.

Response: We have adjusted these figures.

11. Figure 9: the colour scale should be improved, ensuring it is centred on zero and has improved perceptual properties.

Response: Accepted and revised.

12. Figure 10: same as previous figures, using different line styles (dashed, etc...) would make the figure easier to perceive in case of colour vision deficiencies.

**Response:** The modifications have been made, with C-LSAT 2.1 represented by a red solid line and C-LSAT HRv1 by a blue dashed line.

13. Figures 11, 12: ideally, the diverging colormap scales should be symmetrical. **Response:** We have adjusted these figures.

14. Table 5: is the \* denoting statistical significance? Was the uncertainty computed assuming any form of linear dependence or just white (uncorrelated) errors?

**Response:** Thank you for your comment. The \* in Tables 5–6 indicates statistical significance at the 0.05 level, and the explanation has been added to the table captions. The uncertainty was calculated with consideration of the serial correlation in the time series, and we have added relevant explanations. (Lines 230–231)

15. Figure 15: the colour scale should be improved – differences in colour between 20-25  $^{\circ}$ C and 25-30 $^{\circ}$ C are not distinguishable.

**Response:** We have adjusted the colorbar of Figure 15.